# Emergence and Persistent Circulation of Highly Pathogenic Avian Influenza Virus A (H5N8) in Kosovo, May 2021–May 2022

**DOI:** 10.3390/microorganisms11092226

**Published:** 2023-09-02

**Authors:** Armend Cana, Bianca Zecchin, Xhavit Merovci, Alice Fusaro, Edoardo Giussani, Sadik Heta, Kiril Krstevski, Dafina Mehmetukaj, Izedin Goga, Beqe Hulaj, Bafti Murati, Calogero Terregino, Aleksandar Dodovski

**Affiliations:** 1Kosovo Food and Veterinary Agency, Industrial Zone, 10 000 Prishtina, Kosovo; 2UBT—Higher Education Institution, Lagjja Kalabria, 10 000 Prishtina, Kosovo; 3Istituto Zooprofilattico Sperimentale Delle Venezie (IZSVe), 35020 Legnaro, Italy; 4Veterinary Institute, Faculty of Veterinary Medicine in Skopje, Ss. Cyril and Methodius University in Skopje, Lazar Pop Trajkov 5-7, MK-1000 Skopje, North Macedonia; 5Agricultural and Veterinary Faculty, University of Prishtina, Bulevardi Bill Clinton, 10 000 Prishtina, Kosovo

**Keywords:** highly pathogenic avian influenza A H5N8, clade 2.3.4.4b, Kosovo, domestic poultry, phylogenetic network analysis, viruses, persistent circulation

## Abstract

In this study, we report the first outbreak of highly pathogenic avian influenza (HPAI) A H5N8, clade 2.3.4.4b in Kosovo on 19 May 2021. The outbreak consisted of three phases: May–June 2021, September–November 2021, and January–May 2022. In total, 32 backyards and 10 commercial holdings tested positive for the virus. Interestingly, the third and last phase of the outbreak coincided with the massive H5N1 clade 2.3.4.4b epidemic in Europe. Phylogenetic analyses of 28 viral strains from Kosovo revealed that they were closely related to the H5N8 clade 2.3.4.4.b viruses that had been circulating in Albania, Bulgaria, Croatia, Hungary, and Russia in early 2021. Whole genome sequencing of the 25 and partial sequencing of three H5N8 viruses from Kosovo showed high nucleotide identity, forming a distinctive cluster and suggesting a single introduction. The results of the network analysis were in accordance with the three epidemic waves and suggested that the viral diffusion could have been caused by secondary spreads among farms and/or different introductions of the same virus from wild birds. The persistent circulation of the same virus over a one-year period highlights the potential risk of the virus becoming endemic, especially in settings with non-adequate biosecurity.

## 1. Introduction

Avian Influenza (AI) A viruses are a diverse group of viruses causing mild to severe disease in poultry, pigeons, wild birds, and aquatic birds. The natural reservoirs for most low pathogenicity avian influenza (LPAI) viruses are wild birds, particularly birds of wetlands and aquatic environments such as the Anseriformes (particularly ducks, geese, and swans) and Charadriiformes (particularly gulls, terns, and waders) [1]. Highly pathogenic avian influenza (HPAI) viruses, with the potential for severe clinical signs and high mortality rates among poultry, originate from the changes in the hemagglutinin proteolytic cleavage site of H5 or H7 LPAI viruses. HPAI had not been present in wild bird host reservoirs before 2005 [2]. In 1996, a new H5 HPAI strain A/goose/Guangdong/1/1996 (Gs/GD) appeared for the first time in poultry in China, where it circulated for several years and spread to the surrounding countries. However, in 2005, this virus jumped from domestic to wild migratory birds. Since then, the H5 virus of the Gs/GD lineage has spread worldwide, causing mass mortality events in wild birds, great losses in the poultry industry and intermittent human cases [3,4,5]. While co-circulating with LPAI strains, it evolved into multiple genetic clades and variants with significant antigenic diversity [6]. A novel clade 2.3.4.4 reassortant H5N8, known as group A, first appeared in China in 2010, spread through Eurasia, and reached Europe in autumn 2014 [7,8]. In May 2016, an HPAI virus of subtype H5N8, clade 2.3.4.4 of group B (2.3.4.4b) was discovered in waterfowl and shorebirds in Siberia. As of March 2017, the virus had spread across Europe, the Middle East, and Africa [9,10,11,12], causing the most devastating epizootic ever recorded in domestic poultry [13]. Since then, multiple incursions of clade 2.3.4.4b have been reported in Europe in 2019–2020, 2020–2021, and 2021–2022 [14]. Increased deaths among a wide range of wild bird species, which were observed during the 2016–2017 HPAI H5N8 outbreaks in Europe, have shifted the paradigm of wild birds as unaffected agents of HPAI viruses, with increasing concerns about the potential effects on their populations as well as on the biodiversity of the ecosystems [15]. With a total of more than 2300 outbreaks in poultry, 46 million birds culled at the affected premises, and 2700 HPAI occurrences in wild birds throughout 36 European countries, the 2021–2022 HPAI epidemic season has been the largest pandemic to date to be documented in Europe [16]. The virus soon reached the Balkans [13,17,18] through wild bird migratory routes, the Black Sea–Mediterranean Flyway and the Adriatic Flyway. The first outbreak in Kosovo was identified on 19th May, 2021 [19]. Kosovo is a landlocked country in Southeast Europe with an area of 10,887 km^2^. It has a high density of backyard poultry farms as well as a high population of Anseriformes (geese, and domestic ducks), thus representing a favorable ecosystem for the introduction and spread of the infection throughout the country. Kosovo has around 120 commercial poultry farms distributed throughout the country, which host around 1.3 million birds. There are no data on the number of backyard farms. In this study, we report the first outbreak of HPAI A H5N8, clade 2.3.4.4b in Kosovo and the persistent circulation of the virus during the period May 2021–May 2022. Here, we also provide the most common necropsy findings observed, and to investigate the possible transmission dynamics among the affected farms, the complete genome sequences of 25 viruses were subjected to a more in-depth study by using a genetic network analysis.

## 2. Materials and Methods

The sampling frame was part of the passive surveillance annual program of the Kosovo Food and Veterinary Agency for AI and Newcastle disease (ND). Official veterinary inspectors visited all the suspected farms. An epidemiological investigation was carried out. The number and composition of poultry species on the farms and the number of dead poultry at the time of the first visit were recorded. Selected fresh poultry carcasses, as well as tracheal swabs, were submitted to the Kosovo Food and Veterinary Laboratory (KFVL) in the shortest time possible. From the first detected positive H5 Avian influenza case on 19 May 2021 and right up to May 2022, 502 poultry carcasses from 77 backyard farms and 12 commercial farms were submitted to the KFVL. The breakdown by species was as follows: 459 chickens/broilers, 8 turkeys, 22 common quails, 8 domestic ducks, 2 geese, and 3 pigeons. On average, 4.9 carcasses from backyard farms and 9.5 carcasses from commercial farms were collected. In addition, a variable number of oropharyngeal swabs from live poultry were collected from commercial farms only. All animals underwent gross necropsy under appropriate biosafety measures. For further analyses, oropharyngeal and cloacal swabs and tissues such as the lung, trachea, proventriculus, intestine, spleen, and brain were sampled.

### 2.1. Laboratory Analysis

Oropharyngeal swabs from each carcass were processed individually, in 1 X PBS containing 10,000 IU/mL penicillin, 10 mg/mL streptomycin, 0.25 mg/mL gentamicin, and 5000 IU/mL nystatin. Viral RNA from swabs was extracted manually using the QIAamp Viral RNA Mini Kit (Qiagen^®^, Hilden, Germany), according to the manufacturer’s recommendations, whereas extraction from tissues was done manually, starting with the rapid partial disruption of the tissue with PBS supplemented with antibiotics, and continuing with the IndiSpin^®^ Pathogen Kit (Indical Bioscience, Leipzig, Germany). One to two individual tissues/organs representative from each farm were tested. Lungs were the preferred choice, whereas the second choice was proventricle. They were selected based on the presence of extensive lesions. The rest of the organs were kept frozen and stored for further analyses. All samples were tested by real-time RT-PCR for the AI matrix gene [20] and for H5 [21], H7 [22], N1 [23], and N8 [24] genes. Probes and primers were synthesized by Eurofins Genomics (Ebersberg, Germany).

Amplification of target sequences was performed using AgPath-ID One-Step RT-PCR kit (Applied Biosystems™, Waltham, MA, USA) by adding 12.5 μL 2× RT-PCR master mix reagent, 1 μL 25× RT-PCR enzyme mix; the primer and probe concentration for each assay is listed in Appendix A. Five µL of RNA template for AI type A and N1/N8 subtyping or 2 µL of RNA template for H5 and H7 subtyping were used. The 25 µL final reaction volume was adjusted with nuclease-free water up to the final volume for each assay. All reactions were performed on the QuantStudio 5 Real-Time PCR system (Applied Biosystems™, USA). The following thermal profile was used: reverse transcription at 45 °C for 10 min, activation step at 95 °C for 10 min, and 45 amplification cycles with denaturation at 95 °C for 15 s and annealing at 60 °C for AIV type A, and 56 °C for 45 s for H5, H7, N1, and N8. Samples were tested in the same run by using the Veryflex option.

#### 2.1.1. H5 Sanger Sequencing of the Cleavage Site and HPAI Identification

For case definition, selected H5N8-positive samples were submitted to the Faculty of Veterinary Medicine—Skopje, Ss Cyril and Methodius University in Skopje for further characterization of H5 AI virus by One-step RT-PCR and for Sanger sequencing of the hemagglutinin cleavage site according to the protocol [25]. To increase the probability of obtaining sequences of good quality, RNA samples with a high (RNA) viral load were selected. The viral RNA yield in the purified RNA samples was assessed based on the Ct values of the samples obtained with the AIV M-gene RT qPCR.

#### 2.1.2. Whole Genome Sequencing, Phylogenetic and Network Analyses

To confirm the results and further characterize viral strains, HPAI H5N8-positive representative samples from 10 farms sampled in May–June 2021, as well as from 17 farms sampled from September 2021 to April 2022, were sent to the EU Reference Laboratory for Avian Influenza and Newcastle Disease, Istituto Zooprofilattico Sperimentale delle Venezie (IZSVe, Legnaro, Italy). These samples consisted of oropharyngeal swabs and pools of organs such as the lung, trachea, proventriculus, intestine, and spleen from animals based on the low Ct values of the samples obtained previously with the AIV M-gene RT-qPCR. Total RNA was purified by using the QIAamp Viral RNA Mini Kit (Qiagen, Hilden, Germany) according to the manufacturer’s instructions. The SuperScript™ III One-Step RT-PCR System with Plati-num™ Taq High Fidelity DNA Polymerase (Invitrogen, Carlsbard, CA, USA) was used to obtain complete genomes as previously described by Zhou et al., 2009 [26]. The Agencourt AMPure XP (Beckman Coulter Inc., Brea, CA, USA) and Qubit™ DNA HS Assay (Thermo Fisher Scientific, Waltham, MA, USA) were, respectively, used to purify and quantify the amplicons that were then mixed in equimolar proportion. Illumina Nextera XT DNA Sample Preparation Kit (Illumina, San Diego, CA, USA) was used to prepare sequencing libraries. The Illumina MiSeq platform (2 × 250 bp Paired-End; Illumina, San Diego, CA, USA) was employed to perform the sequencing. FastQC v0.11.2 (https://www.bioinformatics.babraham.ac.uk/projects/fastqc/ (accessed on 14 April 2022)) was used to assess the read quality and raw data were filtered by removing reads with more than 100 bases and with a Q score lower than 7, reads with more than 10% of undetermined bases, and duplicated paired-end reads. Illumina Nextera XT adaptors sequences (Illumina, San Diego, CA, USA) were clipped from reads with scythe v0.991 (https://github.com/vsbuffalo/scythe (accessed on 14 April 2022)) and trimmed with sickle v1.33 (https://github.com/najoshi/sickle (accessed on 14 April 2022)). Complete genomes were generated with BWA v0.7.12 (https://github.com/lh3/bwa (accessed on 14 April 2022)) [27] through a reference-based approach and the alignments were processed with Picard-tools v2.1.0 (http://picard.sourceforge.net) and GATK v3.5 (https://github.com/moka-guys/gatk_v3.5 (accessed on 14 April 2022)) [27,28,29]. LoFreq v2.1.2 (https://github.com/CSB5/lofreq (accessed on 14 April 2022)) [30] was used to call Single Nucleotide Polymorphisms (SNPs). Consensus sequences were submitted to the GISAID EpiFlu™ database (http://www.gisaid.org (accessed on 15 April 2022)) under the accession numbers reported in Appendix A. Sequences of each gene segment were aligned in MAFFT v7 [31] and compared to the most related sequences available in GISAID obtained from a BLAST search. Maximum likelihood phylogenetic trees were generated in IQTREE v1.6.6 (https://github.com/iqtree/iqtree1 (accessed on 15 April 2022)) performing an ultrafast bootstrap resampling analysis (with 1000 replications). Phylogenetic trees were visualized in FigTree v1.4.4 (http://tree.bio.ed.ac.uk/software/figtree/ (accessed on 15 April 2022)).

The genetic network was obtained using the Median Joining (MJ) method implemented in NETWORK 10.2.0.0 [32] for the concatenated gene segments of 25 H5N8 non-reassortant viruses from Kosovo (the complete genome was available for 25 out of 28 sequenced viruses, Table 1). The MJ network uses a maximum parsimony approach to reconstruct the relationships between highly similar sequences, displayed by nodes and links connecting the nodes.

### 2.2. Ethical Statement

During the sample collection from live birds, tracheal and cloacal swabs were collected by expert veterinarians in accordance with ethical standards and animal welfare requirements on the Directive of the European Council on the protection of animals used for scientific purposes (2010/63/EU).

## 3. Results

### 3.1. Distribution of Positive Farms

Out of 502 tested carcasses, 177 out of 459 chickens or broilers, 4/8 ducks, 2/2 geese, 2/8 turkeys, and 16/22 common quails were positive for the virus. Tissue samples with positive results on AIV M-gene RT-qPCR showed a lower mean Ct value, with a mean Ct value of 19.78 (n = 58, standard deviation (SD) = 3.84), compared to individual oropharyngeal swabs from carcasses with positive results, with a mean Ct value of 23.80 (n = 201, SD = 4.88).

In the period 19 May 2021–12 May 2022, a total of 32 backyards and 10 commercial farms tested positive for HPAI H5N8 (Figure 1). The outbreak consisted of three phases, i.e., May–June 2021, September–November 2021, and January–May 2022, with the periods in between free of reported cases in poultry. During the first wave (May–June 2021), 20 farms tested positive, 19 of which were backyard farms, and in 11 of them, waterfowl such as geese and domestic ducks were present. The period between the 14 June and 29 September was free of reported cases. The recurrence (September–November 2021) was characterized by the involvement of 7 out of 11 commercial chicken farms. The third wave (January–May 2022) consisted of 11 outbreaks, with 2 out of 11 registered in commercial farms. The overall number of poultry in all the affected farms amounted to 179,198. Breakdown by species was of 175,058 chickens, 492 turkeys, 553 geese, 378 domestic ducks, and 2717 common quails. The number of affected poultry in commercial farms only was of 165,899, representing roughly 12.7% of the entire commercial poultry industry in the country.

### 3.2. Necropsy Finding

The most notable gross lesions at necropsy included: cyanotic comb and legs, multifocal petechiae to ecchymoses of the periventricular mucosa and intestine, and inflammations of the trachea, whereas multifocal necrosis in the pancreas was one of the most striking and consistent features observed during necropsy (Figure 2).

### 3.3. Virus Identification and Pathotyping

Virus isolation in SPF embryonated eggs (according to the Diagnostic Manual Directive 2006/437/CE) and viral intravenous inoculum in SPF chickens (in compliance with the Diagnostic Manual Directive 2006/437/CE)(*) were performed at the EURL—IZSVe on a chicken sample from the first outbreak case. The Intravenous Pathogenicity Index was 3.00.

### 3.4. Phylogenetic and Network Analyses

We characterized either the complete (n = 25) or partial (n = 3) genome of representative HPAI H5N8 viruses collected from backyards (n = 19) and commercial farms (n = 9) during the three epidemic waves. All the viruses from Kosovo belonged to clade 2.3.4.4b and clustered with the viruses that had been circulating in Russia and Europe since the end of 2020; in particular, they showed the highest similarity to the HPAI H5N8 viruses detected in Albania, Bulgaria, Croatia, Hungary, and Russia in early 2021 (Figure 3). The phylogenetic analyses of the eight gene segments (Figure 3, Appendix A) showed that all the HPAI H5N8 viruses from Kosovo cluster together, which suggests one single virus introduction into the country.

To investigate the possible transmission dynamics among the affected farms, the complete genome sequences of 25 viruses were subjected to a more in-depth study by using a genetic network analysis (Figure 4). The viruses showed a clear clustering by epidemic wave. The low number of nucleotide differences between the viruses collected from some of the outbreaks in commercial or backyard farms might be indicative of viral spread among them. In particular, viruses 3124-03, 3124-04, 3124-05, and 3124-07 collected from commercial farms cluster together, showing from 0 to 8 nucleotide differences along the entire genome, and separately from the backyard cases. On the other hand, some of the viruses collected from the backyard farms are closely related to each other, such as viruses 5162-15, 5162-19, 5162-07, 5162-04 (2 to 7 nucleotide differences) or viruses 5162-01, 5162-02, 5162-16 (1 to 5 nucleotide differences), indicating a potential virus spread among them.

## 4. Discussion

In 2020–2021, HPAI H5N8 was the main strain circulating in Europe, which was largely replaced by the HPAI H5N1 strain in the period 2021–2022 [16,19]. Currently, H5N1 has become the most widespread throughout Europe, although it has not been detected in Kosovo so far. Furthermore, Kosovo was one of the few countries where the circulation of HPAI H5N8 persisted for two consecutive epidemics [16,33]. Of particular interest is that the period between the first and second waves, lasting from 14 June to 29 September, despite a very active awareness campaign, was free of reported cases. Part of this period was covered by the strict housing order, which included the closing of all live bird markets and the prohibition of the poultry movement. After the abolishment of the housing order on 13 July 2021, 25 days after the last case during the first phase, the housing order was re-established again on 30 September 2021, after the reoccurrence of AIV H5N8 in a commercial poultry farm on 29 September 2021. The movement of poultry and all markets involving live poultry were prohibited again. All actions were based on Administrative Instruction No. 2005/24 Against Zoonotic Disease Avian Influenza and the National Contingency Plan for AI. The second phase during September raised concern regarding the involvement of a high number of commercial farms. Based on phylogenetic analyses, there is a clear indication of the single introduction of the virus in the country, and it is not possible to exclude a persistent circulation of the virus within the country in the time elapsing between the three epidemic waves in un-sampled hosts (e.g., wild birds). Due to the lack of sequences from wild birds, it is not clear whether the viral diffusion during the outbreak phases was caused by different introductions from wild birds or by secondary spread among farms. The virus could have circulated undetected between phases in resident wild birds or in the backyard poultry sector. However, farm-to-farm transmission of the virus, as suggested by the small number of nucleotide differences among some of the viruses recovered from various outbreaks during the same phase, is also supported by epidemiological investigation. For instance, the farmer of the outbreak on the commercial farm identified as 3124-03 had sold chickens or shared cages with farms identified as 3124-04, 3124-05, and 3124-07, confirming the hypothesis of secondary spread.

To date, no proper surveillance of the wild bird population has been carried out in Kosovo. At this step, it is of crucial importance for the authorities to start implementing active surveillance of bird flu in wild bird populations, as this is important to better evaluate the number of introductions in the poultry farms, as we cannot exclude the circulation in the wild population of highly related viruses. Moreover, surveillance in wild birds can be crucial to determine the possible source of infection for commercial poultry (backyard farms or wild birds, and if wild birds, which species?). This would help to improve surveillance and monitoring strategies. This publication covers the devastating impact that HPAI H5N8 had on the domestic backyard and commercial farms in Kosovo during the three outbreak phases (in 2020–2021 and 2021–2022). Twelve out of 120 chicken commercial farms in the country were affected in the period May 2021–May 2022. They represent around 12.7% of the entire commercial poultry population of the country. The number of backyard farms that may have been affected could be considerably higher than the one officially reported; in particular, those with a small number of poultry. Retrospectively, in identified HPAI H5N8 backyard farm outbreaks, epidemiological investigation revealed that there were sometimes unreported suspected cases within the same village. In fact, backyard farms could have played a significant role in the uncontrolled dissemination of the virus, and are more difficult to control. The persistent circulation of the same virus strain in the period under investigation highlights the potential risk of the virus to become endemic in the future, especially in settings with non-adequate biosecurity [34]. Considering the current pattern of the spread of H5Nx through the Black Sea-Mediterranean Flyway, it is likely that in the future, Kosovo might be exposed to different novel strains of AI. The great number of affected commercial farms highlights the urgent need to improve biosecurity measures and to prevent as much as possible contacts between wild birds and poultry.

The increasing number of reports of mammalian species infected by clade 2.3.4.4b as well as the sporadic human cases reported in recent years highlight the need to constantly monitor the evolution of this virus to promptly detect the emergence of new variants with possible increased zoonotic potential [35,36,37,38,39,40,41,42].

## Figures and Tables

**Figure 1 microorganisms-11-02226-f001:**
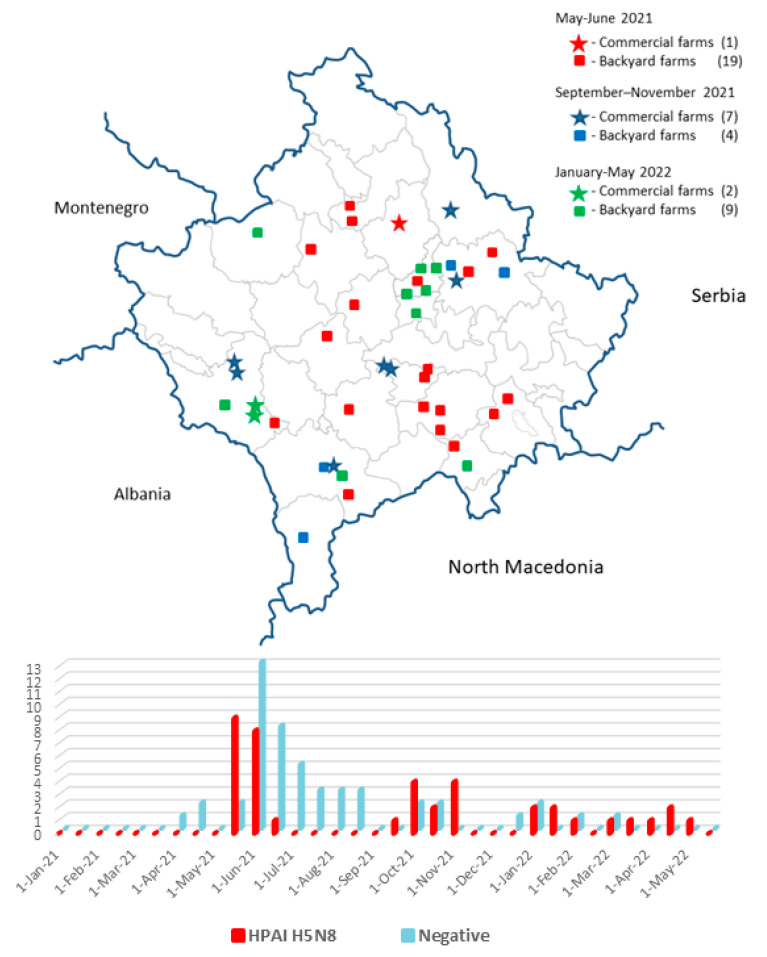
Spatial distribution and incidence timeline of HPAI H5N8-positive farms in Kosovo in the period May 2021–May 2022 (red color), and timeline distribution of suspected AI farms that resulted negative (light blue).

**Figure 2 microorganisms-11-02226-f002:**
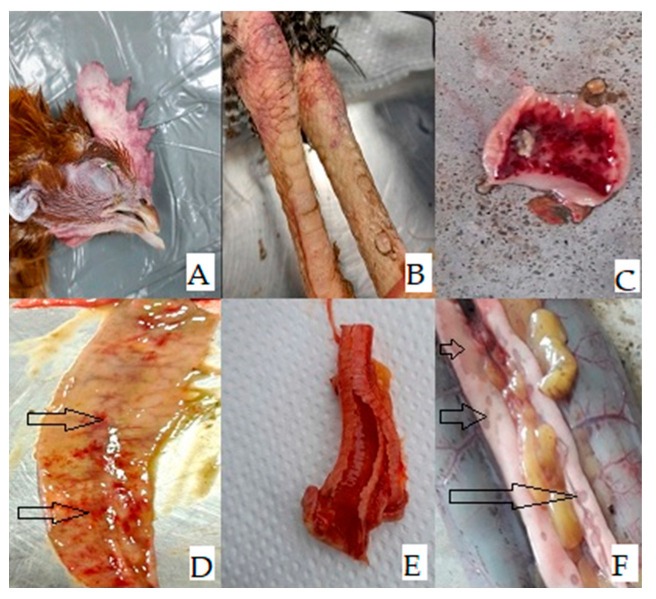
Gross lesions of chicken: (**A**,**B**) Cyanotic comb and legs, (**C**,**D**) Petechiae to ecchymoses of the proventricular mucosa and intestine, (**E**) Inflammations of the trachea and (**F**) multifocal necrosis in the pancreas.

**Figure 3 microorganisms-11-02226-f003:**
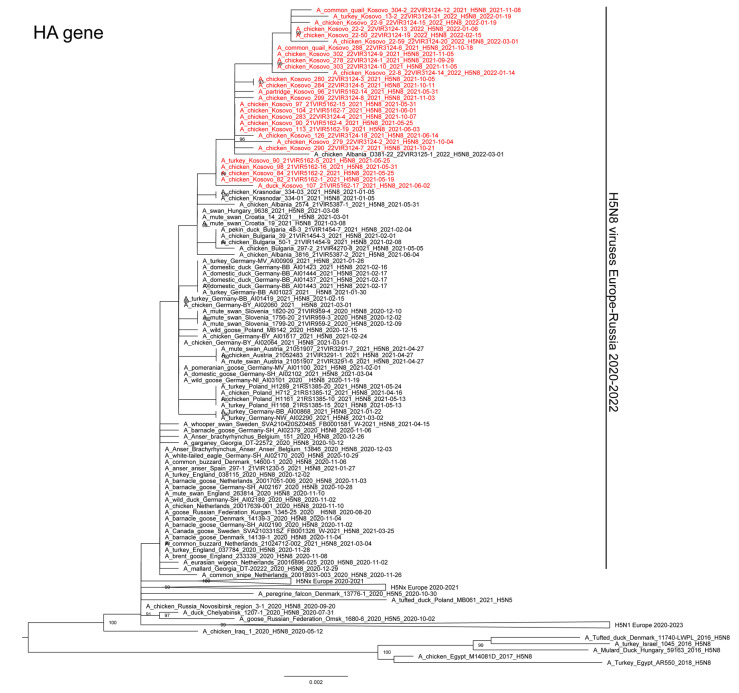
Maximum likelihood phylogenetic tree of the HA gene segment, obtained in IQTREE V1.6.6. Viruses from Kosovo are colored in red; ultrafast bootstrap values higher than 80 are shown next to the nodes.

**Figure 4 microorganisms-11-02226-f004:**
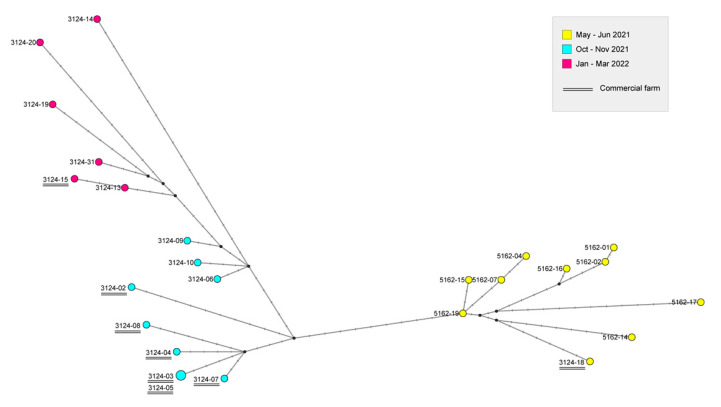
Genetic network generated using the Median Joining method implemented in NETWORK 10.2.0, for the eight concatenated gene segments of 25 non-reassortant H5N8 viruses from Kosovo. Circles represent viral types; the size is proportional to the number of viruses sharing the same type, and the length of the branches is proportional to the number of nucleotide differences. Samples are colored according to the month of collection. Commercial farms are identified by underlined IZSVe-IDs.

**Table 1 microorganisms-11-02226-t001:** HPAI H5N8 viruses from Kosovo sequenced within the framework of the present study.

Virus	District	Category	ID	IZSVe-ID	Date	Genome	Accession Number
A/partridge/Kosovo/96_21VIR5162-14/2021	Prizren	Backyard	96	5162-14	31 May 2021	Complete	EPI_ISL_3128524
A/chicken/Kosovo/97_21VIR5162-15/2021	Gjilan	Backyard	97	5162-15	31 May 2021	Complete	EPI_ISL_3128525
A/chicken/Kosovo/98_21VIR5162-16/2021	Mitrovica	Backyard	98	5162-16	31 May 2021	Complete	EPI_ISL_3128526
A/duck/Kosovo/107_21VIR5162-17/2021	Ferizaj	Backyard	107	5162-17	02 June 2021	Complete	EPI_ISL_3128527
A/chicken/Kosovo/113_21VIR5162-19/2021	Ferizaj	Backyard	113	5162-19	03 June 2021	Complete	EPI_ISL_3128528
A/chicken/Kosovo/82_21VIR5162-1/2021	Mitrovica	Backyard	82	5162-1	19 May 2021	Complete	EPI_ISL_3128529
A/chicken/Kosovo/84_21VIR5162-2/2021	Ferizaj	Backyard	84	5162-2	25 May 2021	Complete	EPI_ISL_3128530
A/chicken/Kosovo/90_21VIR5162-4/2021	Prizren	Backyard	90	5162-4	25 May 2021	Complete	EPI_ISL_3128531
A/chicken/Kosovo/104_21VIR5162-7/2021	Prizren	Backyard	104	5162-7	01 June 2021	Complete	EPI_ISL_3128532
A/chicken/Kosovo/279_22VIR3124-2/2021	Prizren	Commercial	279	3124-2	04 October 2021	Complete	EPI_ISL_12176839
A/chicken/Kosovo/280_22VIR3124-3/2021	Gjakova	Commercial	280	3124-3	05 October 2021	Complete	EPI_ISL_12176840
A/chicken/Kosovo/283_22VIR3124-4/2021	Ferizaj	Commercial	283	3124-4	07 October 2021	Complete	EPI_ISL_12176841
A/chicken/Kosovo/284_22VIR3124-5/2021	Gjakova	Commercial	284	3124-5	11 October 2021	Complete	EPI_ISL_12176843
A/common_quail/Kosovo/288_22VIR3124-6/2021	Prizren	Backyard	288	3124-6	18 October 2021	Complete	EPI_ISL_12176844
A/chicken/Kosovo/290_22VIR3124-7/2021	Pristina	Commercial	290	3124-7	21 October 2021	Complete	EPI_ISL_12176845
A/chicken/Kosovo/299_22VIR3124-8/2021	Ferizaj	Commercial	299	3124-8	03 November 2021	Complete	EPI_ISL_12176846
A/chicken/Kosovo/302_22VIR3124-9/2021	Pristina	Backyard	302	3124-9	05 November 2021	Complete	EPI_ISL_12176847
A/chicken/Kosovo/303_22VIR3124-10/2021	Pristina	Backyard	303	3124-10	05 November 2021	Complete	EPI_ISL_12176848
A/chicken/Kosovo/22-2_22VIR3124-13/2022	Pristina	Backyard	22-2	3124-13	06 January 2022	Complete	EPI_ISL_12176850
A/chicken/Kosovo/22-8_22VIR3124-14/2022	Pristina	Backyard	22-8	3124-14	14 January 2022	Complete	EPI_ISL_12176851
A/chicken/Kosovo/22-9_22VIR3124-15/2022	Gjakova	Commercial	22-9	3124-15	19 January 2022	Complete	EPI_ISL_12176852
A/chicken/Kosovo/126_22VIR3124-18/2021	Mitrovica	Commercial	126	3124-18	14 June 2021	Complete	EPI_ISL_12176853
A/chicken/Kosovo/22-50_22VIR3124-19/2022	Prizren	Backyard	22-50	3124-19	15 February 2022	Complete	EPI_ISL_12176854
A/chicken/Kosovo/22-59_22VIR3124-20/2022	Pristina	Backyard	22-59	3124-20	01 March 2022	Complete	EPI_ISL_12176855
A/turkey/Kosovo/13-2_22VIR3124-31/2022	Prizren	Backyard	13-2	3124-31	19 January 2022	Complete	EPI_ISL_12176931
A/turkey/Kosovo/90_21VIR5162-5/2021	Prizren	Backyard	90	5162-5	25 May 2021	Partial	EPI_ISL_3142407
A/chicken/Kosovo/278_22VIR3124-1/2021	Pristina	Commercial	278	3124-1	29 September 2021	Partial	EPI_ISL_12176838
A/common_quail/Kosovo/304-2_22VIR3124-12/2021	Prizren	Backyard	304-2	3124-12	08 November 2021	Partial	EPI_ISL_12176849

## Data Availability

The consensus sequences of the viruses analysed in this study were submitted to the GISAID EpiFlu™ database under the accession numbers reported in Table 1.

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
