# Peer review of "Emergence and Persistent Circulation of Highly Pathogenic Avian Influenza Virus A (H5N8) in Kosovo, May 2021–May 2022"

_microorganisms, 2023, doi:10.3390/microorganisms11092226_

Round 1

Reviewer 1 Report

This manuscript reported that the H5N8 HPAI viruses in Kosovo, May 2021-May 2022. By sequencing, they obtained the whole genome of 25 viruses and the partial genome of 3 H5N8 viruses. Phylogenetic analysis indicated all the viruses from Kosovo belonged to clade 2.3.4.4b and clustered with the viruses had been circulating in Russia and Europe since the end of 2020. The results of phylogenetic analysis and the network analysis suggested that a persistent circulation of the virus. In general, this study highlights in the future Kosovo might be exposed to different novel strains of AI and enhanced active surveillance in Kosovo is necessary.

Minor issues:

1.      There is no description about the number of samples testing positive for the virus in 502 poultry carcasses. The authors should add concrete figure. And there is no explanation about the principles for the sequencing samples selected from them. Is it randomly selected?

2.      Line209-210. “The low number of nucleotide differences between the viruses…”, the authors should clearly illustrate the specific values of nucleotide identity of the eight gene segments between the viruses.

Reviewer 2 Report

I did the review of the manuscript titled ‘Persistent Circulation of Highly Pathogenic Avian Influenza Virus A(H5N8) in Kosovo, May 2021- May 2022’.

In the study, the authors investigated the first outbreak of highly pathogenic avian influenza (HPAI) A H5N8, clade 2.3.4.4b in Kosovo on May 19, 2021. The outbreak consisted of three temporal phases: May–June 2021, September–November 2021, and January–May 2022. Phylogenetic analyses of 28 viral strains from Kosovo revealed that they were closely related to the H5N8 clade 2.3.4.4.b viruses that have been circulating in Albania, Bulgaria, Croatia, Hungary, and Russia in early 2021.

Below you will find my comments and suggestions:

Materials and Methods

Line 89 to 94: Besides swabs, different tissues were also collected for virus detection. From text, it can be clearly seen that swabs were processed individually. On the other hand, it is quite unclear how the tissue samples were processed – individually or were the tissue samples pooled. If they were pooled, then how? Please explain more detailed.

The sampling strategy for a location should be also explained. How many carcases per location were submitted. How many swabs per location were investigated? Were all investigated swabs taken from live animals or were in some case taken also from dead animals?

Results:

In the M&M section, the number of sampled farms, animals and also different samples collected from carcasses are given. In the Results, only the numbers of positive farms are given. If the authors extend their investigation beyond swab testing, it would may be interesting to analyse these results and present them in the manuscript?

Line 207 to 217 include the results, interpretation of the results and hypothesis. Pleas focus on reporting and analysis of the results in the Results section of the manuscript.

Discussion

This section is not written very well. There are many repetitions of the results, and the result of the study sometimes lacks the interpretation and discussion. This is especially true for the results of phylogenetic and network analyses.

The part between lines 280 and 299 should be omitted or considerably shortened as give the reader well known data that has no ‘connections’ with the present study that did not investigate the impact of H5N8 epidemic in Kosovo in One Health perspective (that will include humans and other susceptible mammals).

Line 274. The reference s missing.

Line 260: there is no doubt that active monitoring of wild birds adds some to clearer epidemiological picture of IAV in reservoir of wild birds. But it may add only a little in a system of poultry infections and virus transmission between farms. I would suggest that authors explain the need for active surveillance or remove this line from the manuscript.

At the end, I would suggest, if possible, to combine the Results and Discussion sections of the manuscript and to focus on the results of the investigation, the analysis of the results, interpretation and the discussion of the data obtained during the investigation.

A minor editing of the text will improve some parts of the manuscript.
